# Immobilization Systems of Antimicrobial Peptide Ib−M1 in Polymeric Nanoparticles Based on Alginate and Chitosan

**DOI:** 10.3390/polym14153149

**Published:** 2022-08-02

**Authors:** Carlos Enrique Osorio-Alvarado, Jose Luis Ropero-Vega, Ana Elvira Farfán-García, Johanna Marcela Flórez-Castillo

**Affiliations:** 1Universidad de Santander, Facultad de Ciencias Naturales, Ciencias Básicas y Aplicadas para la Sostenibilidad-CIBAS, Calle 70 No. 55-210, Bucaramanga 680003, Colombia; jose.ropero@udes.edu.co; 2Universidad de Santander, Facultad de Ciencias Médicas y de la Salud, Instituto de Investigación Masira, Calle 70 No. 55-210, Bucaramanga 680003, Colombia; afarfan@udes.edu.co

**Keywords:** nanoparticles, alginate, chitosan, Ib-M peptides, *E. coli*, peptide stability

## Abstract

The development of new strategies to reduce the use of traditional antibiotics has been a topic of global interest due to the resistance generated by multiresistant microorganisms, including *Escherichia coli*, as etiological agents of various diseases. Antimicrobial peptides are presented as an alternative for the treatment of infectious diseases caused by this type of microorganism. The Ib−M1 peptide meets the requirements to be used as an antimicrobial compound. However, it is necessary to use strategies that generate protection and resist the conditions encountered in a biological system. Therefore, in this study, we synthesized alginate and chitosan nanoparticles (Alg−Chi NPs) using the ionic gelation technique, which allows for the crosslinking of polymeric chains arranged in nanostructures by intermolecular interactions that can be either covalent or non-covalent. Such interactions can be achieved through the use of crosslinking agents that facilitate this binding. This technique allows for immobilization of the Ib−M1 peptide to form an Ib−M1/Alg−Chi bioconjugate. SEM, DLS, and FT-IR were used to determine the structural features of the nanoparticles. We evaluated the biological activity against *E. coli* ATCC 25922 and Vero mammalian cells, as well as the stability at various temperatures, pH, and proteases, of Ib−M1 and Ib−M1/Alg-Chi. The results showed agglomerates of nanoparticles with average sizes of 150 nm; an MIC of 12.5 µM, which was maintained in the bioconjugate; and cytotoxicity values close to 40%. Stability was maintained against pH and temperature; in proteases, it was only evidenced against pepsin in Ib−M1/Alg-Chi. The results are promising with respect to the use of Ib−M1 and Ib−M1/Alg−Chi as possible antimicrobial agents.

## 1. Introduction

Owing to their biocompatibility, biodegradability, bioavailability, and low-toxicity properties, biopolymers, such as alginate and chitosan, are widely used in the biomedical and pharmaceutical industries [1]. They have a wide range of medical applications in tissue engineering, implants, and drug delivery [2,3,4,5]. Additionally, they have reactive functional groups that allow for the conjugation of peptides and proteins. [5,6,7,8,9,10].

Different strategies have been used to explore the preparation of polymeric nanoparticles, including coprecipitation, chemical crosslinking, thermodecomposition, coacervation, emulsification, and ionic gelation [11,12,13,14]. The latter is widely used for the preparation of nanoparticles of alginate and chitosan with the aim of bioconjugating them with antimicrobial peptides [7,15].

Antimicrobial peptides (AMPs) are small molecules composed of a length of 12 to 50 amino acids and are usually positively charged and amphiphilic [16]. They are among the body’s first line of defense against the inactivation of pathogens, such as Gram-negative and Gram-positive bacteria, fungi, viruses, and parasites. The positive charge of AMPs allows for initial binding to the membrane via electrostatic interaction [17].

Despite their high antimicrobial activity, most AMPs are not widely used in clinical settings due to limitations such as toxicity and stability. Thus, AMPs present considerable challenges when considering the type of administration. For example, in oral administration, the pH and proteases in the gastrointestinal tract can inhibit the action of AMPs by hydrolysis or denaturation [18]. The conjugation of AMPs with nanoparticles (NPs) has been proposed to increase the local concentration of the peptide and improve its antimicrobial activity [19,20,21,22]. Alginate and chitosan nanoparticles (Alg−Chi NPs) have been used to achieve such strategies, as they preserve their structure and can therefore enhance bioactivity [14].

Chitosan is a linear polysaccharide composed of D-glucosamine and N-acetyl-D-glucosamine. It is a deacetylated form of chitin, the structural component of the exoskeletons of crustaceans [23]. The amino (-NH2) and hydroxyl (-OH) groups of the polymer guarantee high reactivity and charge, allowing for its union with different biomolecules [23]. In addition, alginate is a non-toxic polysaccharide composed of one to four linked β-D-mannuronate and α-l-guluronate blocks and can form chitosan-crosslinked gels. Owing to its properties, it is qualified as one of the best drug delivery systems and bioapplications, particularly in the immobilization of peptides, affording bioconjugates with equal or improved biological activity [24,25,26].

The Ib−M1 peptide is part of the group of Ib-M peptides synthesized by Flórez-Castillo et al. [27], which exhibit very good antimicrobial activity at low inhibitory concentrations against clinical and reference strains of *Escherichia coli* [27,28,29]. Ib−M1 peptides have a net charge of +6 and an isoelectric point of 12.5. Ib-M peptides are promising for clinical applications [28,29,30].

In this study, we immobilized the Ib−M1 peptide on polymeric nanoparticles of alginate and chitosan to maintain the antibacterial activity of this peptide and increase its stability with respect to proteases, changes in pH, and temperature. *E. coli* ATCC 25922 was used as a reference microorganism to evaluate antibacterial activity. In this experimental study, we report, for the first time, the effect of the immobilization of the Ib−M1 peptide on Alg−Chi nanoparticles to establish the scope of its clinical use.

## 2. Materials and Methods

### 2.1. Materials

Peptide Ib−M1 (sequence EWGRRMMGRGPGRRMMRWWR-NH2 [27]) was purchased through the commercial company Biomatik USA (Wilmington, DE, USA). Chitosan (Sigma-Aldrich, St. Louis, MO, USA, ≥75%), alginate (Sigma-Aldrich, St. Louis, MO, USA), sodium tripolyphosphate (TPP, Sigma-Aldrich, St. Louis, MO, USA), acetic acid (CH₃COOH, Merck KGaA, Darmstadt, Germany), 2-(1H-Benzotriazole-1-yl)-1,1,3,3-tetraethylammonium tetrafluoroborate (TBTU, Sigma-Aldrich, St. Louis, MO, USA), N, N-diisopropylethylamine (DIPEA, Sigma-Aldrich, St. Louis, MO, USA); *Escherichia coli* strain ATCC 25922, Vero ATCC CCL-81 cell line, Müller–Hinton broth (MH—Scharlau), Luria–Bertani broth (LB Broth, Oxoid, Basingstoke, England), trypsin, phosphate-buffered saline (PBS), dimethyl sulfoxide (DMSO, Sigma-Aldrich, St. Louis, MO, USA), bromide-3 (4,5-dimethylthiazole- 2-yl) 2,5-diphenyltetrazolium (MTT), streptomycin, Dulbecco’s Modified Eagle Medium (DMEM), fetal bovine serum. 

### 2.2. Preparation of Alginate-Chitosan Nanoparticles (Alg−Chi NPs)

Alg−Chi NPs were prepared by the ionic gelation method using pentasodium tripolyphosphate (TPP) as a crosslinking agent [31,32]. A 0.1% *w*/*v* chitosan solution (1% acetic acid and pH 3.5) and aqueous solutions of 1% *w*/*v* alginate and 1% *w*/*v* TPP were prepared independently. Subsequently, 1 mL of the TPP solution was added dropwise and under constant stirring to 9 mL of the chitosan solution. The obtained suspension was stirred for 2 h; then, 10 mL of type I water was added. Immediately afterward, 20 mL of the alginate solution was added dropwise and under constant stirring. The obtained suspension was left to stand for 18 h. Finally, the suspension was agitated for 30 min at 1500 rpm and centrifuged for 10 min at 5000 rpm to remove larger agglomerates. The supernatant was stored for characterization tests.

### 2.3. Structural Characterization of Alg−Chi Nanoparticles

Scanning electron microscopy (SEM) was used to explore the morphology of the Alg−Chi NPs with a Quanta field emission gun (Model 650) operated at 20.0 kV. Images were obtained in secondary electron mode.

To determine the size of the Alg−Chi obtained NPs, dynamic light scattering (DLS) measurements were used with a Zetasizer ZS90 (Malvern) equipped with a helium-neon gas laser with a wavelength of 632 nm.

Fourier transforms infrared (FTIR) measurements were used to verify the presence of the polymers in the synthesized Alg−Chi NPs, which identified the presence of the functional groups of each compound. A Bruker Tensor II spectrometer equipped with a platinum ATR cell and a cooled deuterated triglycine sulfate (DTGS) detector were used.

### 2.4. Preparation of Ib−M1/Alg−Chi Bioconjugate

The Ib−M1 peptide in the Alg−Chi NPs was immobilized according to the methodology of Ropero-Vega [33]. Briefly, the carboxyl group in the Glu-1 residue of the peptide was activated. To this end, 44.2 mg of TBTU and 24.2 µL of DIPEA were added to a 2000 µM solution of Ib−M1 in Tris HCl buffer (10 mM pH 7.4), allowing the reaction to take place for 20 min under constant stirring. Subsequently, this reaction mixture was added to 0.4 mg/mL of nanoparticles and left under constant stirring for 2 h, allowing for the formation reaction of the peptide bond between the activated carboxyl group and the amino groups of the nanoparticles. Finally, the Ib−M1/Alg−Chi bioconjugate was separated by centrifugation at 15,000 rpm for 45 min at 18 °C.

To determine the amount of immobilized peptide, the concentration of peptide in the supernatant was determined using the Bradford method [34] with a Quick Start™ Bradford protein assay kit from BioRad at 595 nm. Before the preparation of the immobilized peptide, the free peptide solution was measured under the same conditions. With these data and using the following formula, it was possible to determine the amount of peptide immobilized in the nanoparticles forming the Ib-M/Alg−Chi bioconjugate:$$\%immobilization=\frac{Free\;peptide\;abs-supernatant\;abs}{Free\;peptide\;abs}*100$$
where Free peptide abs is the absorbance of the peptide activated with TBTU and DIPEA before being combined with the nanoparticles for immobilization, and supernatant abs is the absorbance of the supernatant from the first centrifugation [33].

### 2.5. Antimicrobial Activity against E. coli ATCC 25922

*Escherichia coli* strain ATCC 25922 was cyropreserved at −80 °C in Luria–Bertani broth (LBB) with 15% glycerol. For the reactivation of the microorganism, 50 μL of cryopreserved material was added to 5 mL of LB and incubated at 35 ± 2 °C for 18 to 24 h before each assay.

The minimum inhibitory concentration (MIC) was determined using the microdilution method as described in the Clinical and Laboratory Standards Institute protocol M07A9 [35]. Briefly, twofold serial dilutions of the Ib−M1 peptide (100 and 0.78 μM concentration) were incubated in 96-well, round-bottom plates at 200 μL/well final volume for 24 h at 37 °C with shaking at 150 rpm with 5 × 10^5^ colony forming units per mL (CFU/mL) of bacterial inoculum. The bacterial suspension was adjusted from a concentration of 1 × 10^6^ CFU/mL. Absorbance at 595 nm was determined every hour for 8 h, with a final measurement at 24 h with a microplate reader kit (Bio-Rad, iMark). Mueller–Hinton (MH) broth and *E. coli* in MH broth were taken as negative and growth positive controls, respectively. The MIC was defined as the lowest peptide concentration that inhibited bacterial growth after 24 h. Data represent at least two independent experiments.

### 2.6. Cytotoxicity of Ib−M1 Peptides and Ib−M1/Alg−Chi Bioconjugate

The indirect cytotoxicity of the membranes against Vero cell cultures was analyzed using the MTT (3-(4,5-Dimethyl-2-thiazolyl)-2,5-diphenyl-2H-tetrazolium bromide) reduction method. VERO cells at a concentration of 3 × 10^4^ cells/mL were cultured in 96-well, flat-bottom plates with DMEM medium supplemented with 10% inactivated fetal bovine serum and incubated at 37 °C in a 5% CO_2_ atmosphere until reaching a confluence greater than 90%. Cells were then exposed to serial 1:2 dilutions of peptide Ib−M1 at concentrations in the range of 200 μM to 0.78 μM, with concentrations of the MIC and 1/2 MIC in the bioconjugates and the Alg−Chi NPs at 0.4 mg/mL for 24 h. Subsequently, 20 μL of MTT in PBS pH 7.4 was added to each well at a concentration of 5 mg/mL and incubated for 4 more hours under the same conditions described above, after which time the culture medium was removed from the wells, and 100 μL/well of DMSO was added to solubilize the formazan crystals, which were measured by the absorbances obtained in the spectrophotometric readings at 570 nm to calculate the percentage of cytotoxicity of the compounds. Cells in culture medium receiving no treatment were employed as a negative control.

### 2.7. Stability of Peptide Ib−M1 and Ib−M1/Alg−Chi Bioconjugate

The stability of the free peptide and the Ib−M1/Alg−Chi bioconjugate was determined under the conditions listed in Table 1. Subsequently, the antimicrobial activity against *E. coli* was determined, as mentioned in the previous section.

#### 2.7.1. pH Stability

The peptide and the bioconjugate were dissolved in glycine buffer solution at pH 2 and pH 11 for stability tests. These solutions were left for 30 min at 37 °C under stirring at 120 rpm; then, the pH was adjusted to 7, and the antimicrobial activity was determined as indicated in Section 2.5 [36].

#### 2.7.2. Temperature Stability

For the stability tests at various temperatures, the Ib−M1 peptide and the Ib−M1/Alg-Chi bioconjugate were left for 90 min at the indicated temperatures. These tests were carried out on agitation plates. Then, antimicrobial activity was determined, as previously mentioned in Section 2.5 [37].

#### 2.7.3. Trypsin and Pepsin Stability

Ib−M1 peptide and Ib−M1/Alg−Chi bioconjugate were exposed to pepsin and trypsin in a ratio (enzyme: peptide) of 27:1 and 1:20, respectively, in a beaker. This solution was stirred at 120 rpm at 37 °C for 90 min, taking a sample every 30 min, starting with an initial sample at 0 min. For an activity with pepsin, it was necessary to adjust the pH of the solution to 2 in order to simulate gastric conditions and allow for the action of the enzyme [38]. Assays with bioconjugates were performed in the same way as for a free peptide, adjusting concentrations to maintain the enzyme: peptide ratio. Then, antimicrobial activity was determined, as previously mentioned in Section 2.5.

### 2.8. Statistical Analysis

Except where indicated otherwise, all results are representative of two experiments, and each experiment was replicated three times. Arithmetic means values ± standard deviations are reported for each case. All analyses and graphics were generated using Origin (Pro) version 2019 (OriginLab Corporation, Northampton, MA, USA). 

## 3. Results

### 3.1. Preparation and Characterization of Alg−Chi NPs

Figure 1 shows preparation scheme of Alg−Chi NPs. TPP was used as a crosslinking agent due to its high negative charge, allowing it to interact with the amino groups of chitosan. Similarly, the negative charge of the alginate interacts with the positive surface charges of the chitosan-TPP aggregates, allowing for the formation of the Alg−Chi NPs.

The obtained NPs present with high agglomeration, as evidenced by the SEM micrograph presented in the inset of Figure 1. This aggregation is responsible for the size of the nanoparticles, which was determined to be 134.6 nm by DLS. Despite this aggregation, the dispersion index of the nanoparticle suspension corresponded to 0.177, which indicates that the obtained size distribution is homogeneous.

The Figure 2 shows the FTIR spectrum of Alg−Chi NPs; the stretching presented at 3267 cm^−1^ corresponds to the O-H and N-H groups of the polymers that are present in their structures. The band at 2916 cm^−1^ is attribuided to C-H strectching vibrations. The band at 1589 cm^−1^ corresponds to the carbonyl group of Alg [39], whereas the band at 1430 cm^−1^ is assigned to the amino group of chitosan. The intense band that appears at 1040 cm^−1^ is associated with molecules present in the polymers. Bands associated with TPP can also be seen in the spectrum; for example, the band at 1290 cm^−1^ is associated with *p* = C strectching, that at 1084 cm^−1^ is associated with symmetric and antisymmetric stretching vibrations in the PO_2_ group, that at 1020 cm^−1^ is associated with symmetric and antisymmetric stretching vibrations in the PO_3_ group, and that at 820 cm^−1^ corresponds to antisymmetric stretching of the P-O-P bridge [40].

### 3.2. Preparation of the Ib−M1/Alg−Chi Bioconjugate

Figure 3A shows the reaction scheme for the immobilization of the Ib−M2 peptide on the surface of Alg−Chi NPs. 

An amide bond was formed between the carboxyl group of Glu−1 of the peptide and the amino groups on the surface of the NPs. Immobilization was monitored by determining free amino groups in the suspension using the Bradford method. As shown in Figure 3B, after 2 h of reaction, 37% immobilization of the peptide on the Alg−Chi NPs was reached.

### 3.3. Antimicrobial Activity against E. coli 

The antimicrobial activity of peptide Ib−M1 against *E. coli* 25922 was determined. To this end, the MIC was determined using the microdilution method, with streptomycin as the reference antibiotic. Figure 4 shows the growth kinetics of free Ib−M1 in the presence of streptomycin at 6.25 µM and peptide Ib−M1 at concentrations between 25 and 3.12 µM.

As shown in Figure 4, the minimum inhibitory concentration of Ib−M1 was 12.5 µM, which is concentration at which the growth of *E. coli* is inhibited for up to 24 h.

The MIC of the Ib−M1/Alg−Chi bioconjugate was determined by incubation with *E. coli* 25922 at an immobilized peptide concentration of between 12.5 and 0.78 µM. Figure 5 shows the growth kinetics of the bacteria in the presence of the Ib−M1/Alg−Chi bioconjugate and Alg−Chi NPs at a concentration of 0.4 mg/mL.

As shown in Figure 5, the peptide activity was maintained when it was immobilized on the nanoparticles. In addition, a synergistic effect was observed in the combination of the Alg−Chi NPs with the peptide. Figure 5 shows that Ib−M1 immobilized at a concentration of 6.25 µM inhibited approximately 35% of the growth of the microorganism after 24 h of incubation with the free peptide in the same concentration (Figure 4). On the other hand, the nanoparticles did not affect the growth of the bacteria. Therefore, Alg and chitosan did not show antimicrobial activity against *E. coli*.

### 3.4. Cytotoxicity

Figure 6 shows the percentage of cytotoxicity of Ib−M1 and Ib−M1/Alg-Chit at the MIC determined for *E. coli* strain 25922 (12.5 µM); the cytotoxicity of the Alg−Chi NPs (0.4 mg/mL) was also recorded.

The free peptide at a concentration of 12.5 µM (MIC) exhibited an approximate cytotoxicity of 5%, showing a statistically significant difference with respect to the Ib−M1/Alg−Chi bioconjugate, with a *p* value < 0.0001 (42.19 ± 10.13). The cytotoxicity value of the Alg−Chi NPs also showed a statistically significant difference (*p* ≤ 0.0001) with respect to the free Ib−M1 peptide.

### 3.5. Stability of Ib−M1 and Ib−M1/Alg−Chit

#### 3.5.1. pH Stability

Figure 7 shows the results of the antimicrobial activity with prior exposure to different pH conditions. As shown in Figure 5, the activity of the Ib−M1 peptide and the Ib−M1/Alg−Chi biconjugate was not affected under alkaline and acid conditions.

#### 3.5.2. Thermal Stability

The influence of the temperatures used in the tests on Ib−M1 and Ib−M1/Alg−Chi are shown in Figure 8. There was no decrease in the activity of the compounds, which indicates good stability of the peptide and the bioconjugate during thermal treatments.

#### 3.5.3. Proteolytic Stability 

The effect of the proteolytic activity of trypsin and pepsin on the free peptide and the bioconjugate is shown Figure 9A,B, respectively.

The free peptide Ib−M1, as well as the immobilized peptide Ib−M1/Alg-Chi, showed instability against the proteolytic action of trypsin, with proliferation of the microorganism at 24 h and optical densities like that of the growth control group. However, a decrease in the microbial population was observed at 24 h with the Ib−M1/Alg−Chi bioconjugate, and the growth of the microorganism slowed down within the first 8 h for the same wells, whereas the stability of the free peptide against pepsin was lost once it was exposed to this protease. On the contrary, the Ib−M1/Alg−Chi bioconjugate maintained its antimicrobial activity after being exposed to pepsin.

## 4. Discussion

Results obtained in this work are consistent with those reported by Bagre et al. [41]. Using SEM, they observed that alginate and chitosan nanoparticles obtained by ionic gelation were agglomerated. Furthermore, they observed particle sizes of 213 ± 3.8 nm, whereas in the present study, we obtained particle sizes of 150 ± 56 nm with a polydispersity value of 0.177, which indicates particles of homogeneous size. This aggregation is due to the tight crosslinking caused by the TPP with the chitosan molecules [41].

We prepared bioconjugates according to the methodology described by Ropero-Vega et al. [33], who used iron oxide nanoparticles coated with chitosan to immobilize the Ib-M2 peptide, obtaining a value of 30% of the immobilized peptide. In the present study, we achieved 37% immobilization, a value close to that reported by Ropero-Vega et al. The above shows that this method allows for the immobilization of Ib-M peptides on chitosan, which could expand the applications to obtaining antibacterial surfaces based on this polymer.

*E. coli* (ATCC 25922) was used as a reference microorganism for antimicrobial activity and was inhibited for the free peptide Ib−M1 at 12.5 µM. The result shows promising characteristics with respect to the clinical use of peptides to treat infectious diseases caused by microbial agents such as *E. coli.* Prada et al. [28] used the Ib−M1 peptide as an antimicrobial from different species of *E. coli* as ATCC reference strains and clinical isolates with MIC values of 4.7 and 1.6 µM, respectively. In addition, Flórez-Castillo et al. [30] showed the antimicrobial action of Ib−M1 against fourteen strains of pathogenic *E. coli*. The characteristics of the Ib-M peptide have been associated mainly with its positive charge and increased hydrophobicity as a result of the presence of arginine and tryptophan residues in the sequence [27].

The antimicrobial activity of the Ib−M1 peptide was not affected by immobilization in the Alg−Chi NPs. The above was evidenced by the MIC being maintained in the bioconjugate. These immobilization strategies have been used in various ways and have the advantage of preserving the properties of the compound of interest. Additionally, it was found that Alg−Chi NPs do not show antibacterial activity against *E. coli* at the evaluated concentrations, showing that the activity of the bioconjugate is due only to the presence of the peptide. This result is significant because it shows that the immobilization of the Ib−M1 peptide does not affect its activity. In addition, the activity of the peptide immobilized at 0.5 × CMI (6.25 µM) prolonged the latent phase of the bacteria for at least 8 h and reduced its final concentration by up to 40% at 24 h relative to the growth control. These properties are similar to those reported in previous studies, wherein Ib-M2 was immobilized on iron oxide nanoparticles and its antibacterial activity was maintained against *E. coli* O157:H7 [33]. 

The cytotoxic effect of Ib−M1 and Ib−M1/Alg−Chi on Vero cells was evaluated by MTT assay. Our studies showed that Ib−M1 at the concentration of the CMI (12.5 µM) did not result in significant percentages of cytotoxicity, whereas the cytotoxicity generated by the bioconjugate showed a considerable increase, as well as the nanoparticles without the peptide. This result is consistent with that reported by Prada-Prada et al. [28], who observed that the Ib−M1 peptide presented cytotoxic concentration 50 (CC_50_) in Vero cells at 395.2 µM. These results differ from those reported in previous studies with Alg−Chi polymers in the Vero cell line, as we observed 10% cytotoxicity (cell viability of 90%) [42]. Alginate and chitosan have been used in studies aimed at drug delivery and as healing polymers for skin wounds owing to their biocompatible properties, with low cytotoxicity in Vero cell lines and BHK-21 cells [43].

The stability tests at pH 2 and 11 during the 30 min of treatment and the thermal tests (4 and 100 °C) revealed that the Ib−M1 peptide and the Ib−M1/Alg−Chi bioconjugate maintained stable antimicrobial activity under these conditions. In contrast, Fahimirad et al. [44], evaluated the free antimicrobial peptide LL37 and observed that acidic pH (2) inhibits antibacterial activity. However, at neutral and basic pH, its activity is maintained, and when boiled for longer than 10 min, it loses stability, increasing the MIC. However, these characteristics improved once the LL37 peptide was immobilized in Chi NPs, maintaining its antimicrobial activity in terms of MIC under the evaluated pH and temperature conditions. These findings are also consistent with those reported by Yu et al., who showed higher stability of the antimicrobial peptide microcin J25 (MccJ25) when conjugated with chitosan nanoparticles [45]. Therefore, the stability of the Ib−M1 peptide is another characteristic that makes it promising for use as a new antimicrobial compound.

Immobilization of the peptide did not prevent the effect of trypsin; therefore, this protease can break peptide bonds, affecting the free and immobilized peptide activity. Trypsin is cleaved into amino acids adjacent to Arg and Lys [46]. The structure of the Ib−M1 peptide is rich in arginine; about 35% of the polypeptide chain is composed of this amino acid. The peptide is easily hydrolyzed upon exposure to trypsin, even when the peptide is immobilized in the NPs, generating oligopeptide chains with no activity against *E. coli*.

Pepsin is a protease that preferentially cleaves peptide bonds of amino acids, such as Leu, Ile, Phe, Val, and Trp [47]. Ib−M1 contains three Trp residues (W) susceptible to hydrolysis by pepsin. Therefore, the activity of the Ib−M1 peptide was affected in the presence of this protease. However, this effect was reduced in the Ib−M1/Alg−Chi bioconjugate. It is possible that the immobilization of the peptide makes it difficult for pepsin to access the Trp residues of the peptide.

## 5. Conclusions

Alg−Chi NPs was synthesized by the ionic gelation method. However, due to the properties of the polymers, agglomerates with average sizes of 150 nm were observed. Bioconjugates were obtained by forming a bond between the amino group of chitosan and the carboxyl group of glutamic acid of Ib−M1. 

Peptide Ib−M1 was found to be resistant to temperature and pH conditions. Furthermore, its immobilization in nanoparticles did not affect its activity against *E. coli.* On the contrary, it maintained its characteristics and achieved stability when the peptide was exposed to pepsin, maintaining its antimicrobial properties at MIC concentrations.

The Ib−M1 peptide exhibited low cytotoxic percentages against Vero cells, again showing this compound’s favorable characteristics for biomedical applications. On the other hand, cytotoxicity percentages of 40% were observed for the Ib−M1/Alg−Chi bioconjugate. These values are attributed to Alg−Chi NPs and can be explained by their size. Therefore, further studies are necessary to reduce aggregation and obtain smaller particle sizes.

## Figures and Tables

**Figure 1 polymers-14-03149-f001:**
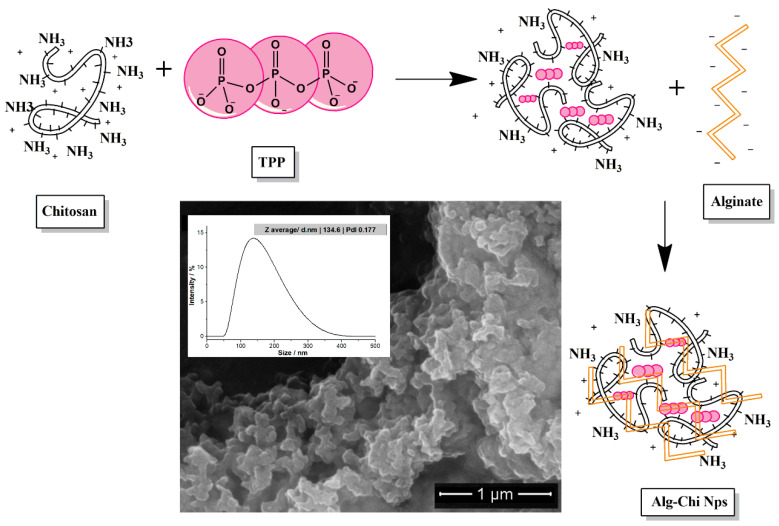
Scheme: synthesis of Alg-Chit NPs by ionic gelation. Inset: scanning electron micrograph of Alg−Chi NPs at 40.000X and size distribution of Alg−Chi NPs. Alg−Chit NPs: alginate-chitosan nanoparticles.

**Figure 2 polymers-14-03149-f002:**
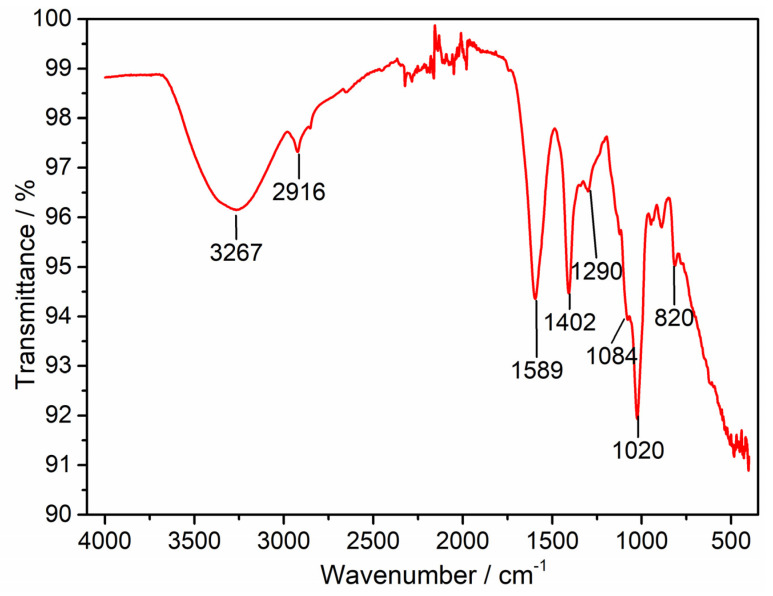
FTIR spectrum of Alg−Chi NPs. FTIR: Fourier transform infrared spectroscopy.

**Figure 3 polymers-14-03149-f003:**
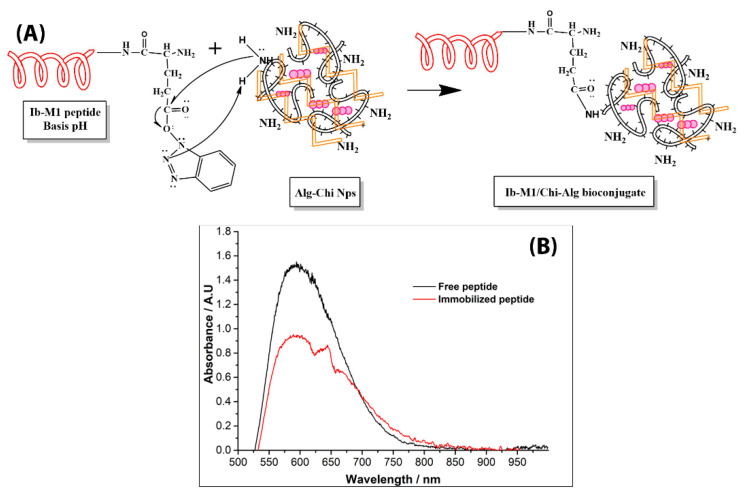
(**A**) Reaction scheme into amino groups of chitosan of Alg−Chi NPs and the carboxyl group of Glu residue of Ib−M1. (**B**) Monitoring of immobilization of the peptide by UV–Vis. Alg−Chit NPs: alginate-chitosan nanoparticles.

**Figure 4 polymers-14-03149-f004:**
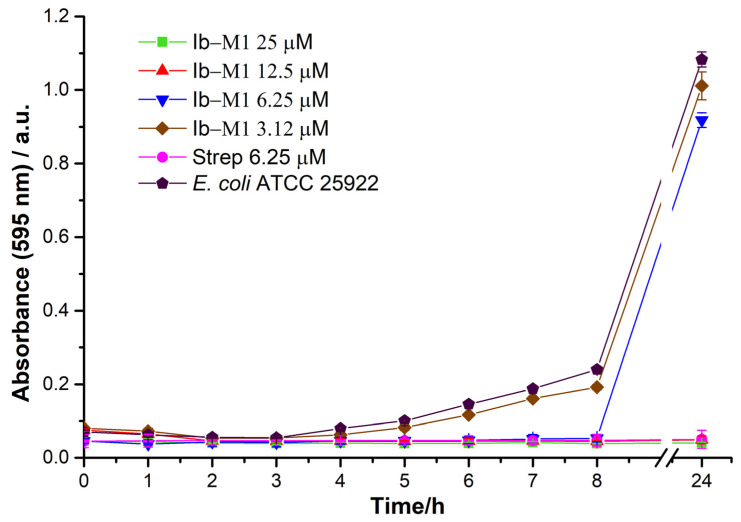
Growth kinetics of *E. coli* (ATCC 25922) in the presence of free Ib−M1 peptide at 25 µM (■), 12.5 µM (▲), 6.25 µM (▼), and 3.12 µM (◆), with streptomycin as reference antibiotic at 6.25 µM (⬤). Growth of *E. coli* without the addition of compounds (⬟) was included for comparison purposes. Each result was evaluated in triplicate in two independent experiments and expressed in terms of arithmetic average plus standard deviation. The Ib-M1 concentrations evaluated ranged from 100 µM to 0.78 µM; and are plotted in the Appendix A.

**Figure 5 polymers-14-03149-f005:**
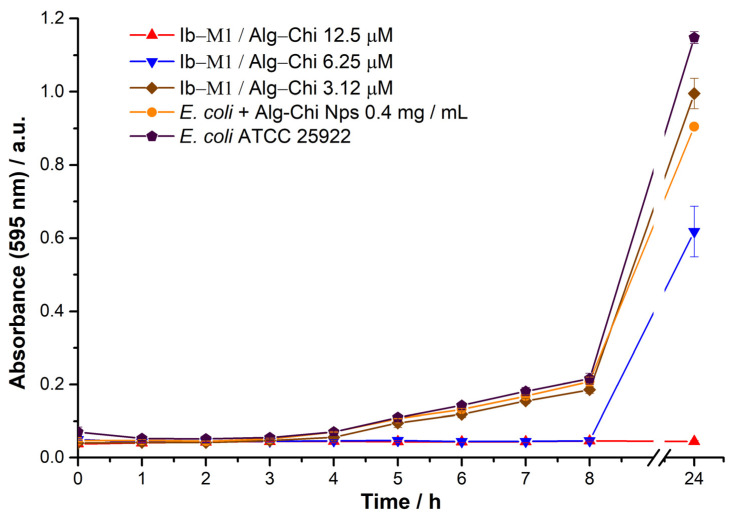
Growth kinetics of *E. coli* (ATCC 25922) in the presence of the Ib−M1/Alg−Chi bioconjugate at 12.5 µM (▲), 6.25 µM (▼), and 3.12 µM (◆). Results from *E. coli* with Alg−Chi NPs at 0.4 mg/mL (⬤) and *E. coli* without the addition of compounds (⬟) are included for comparison purposes. Each result was evaluated in triplicate in two independent experiments and expressed in terms of arithmetic average plus standard deviation.

**Figure 6 polymers-14-03149-f006:**
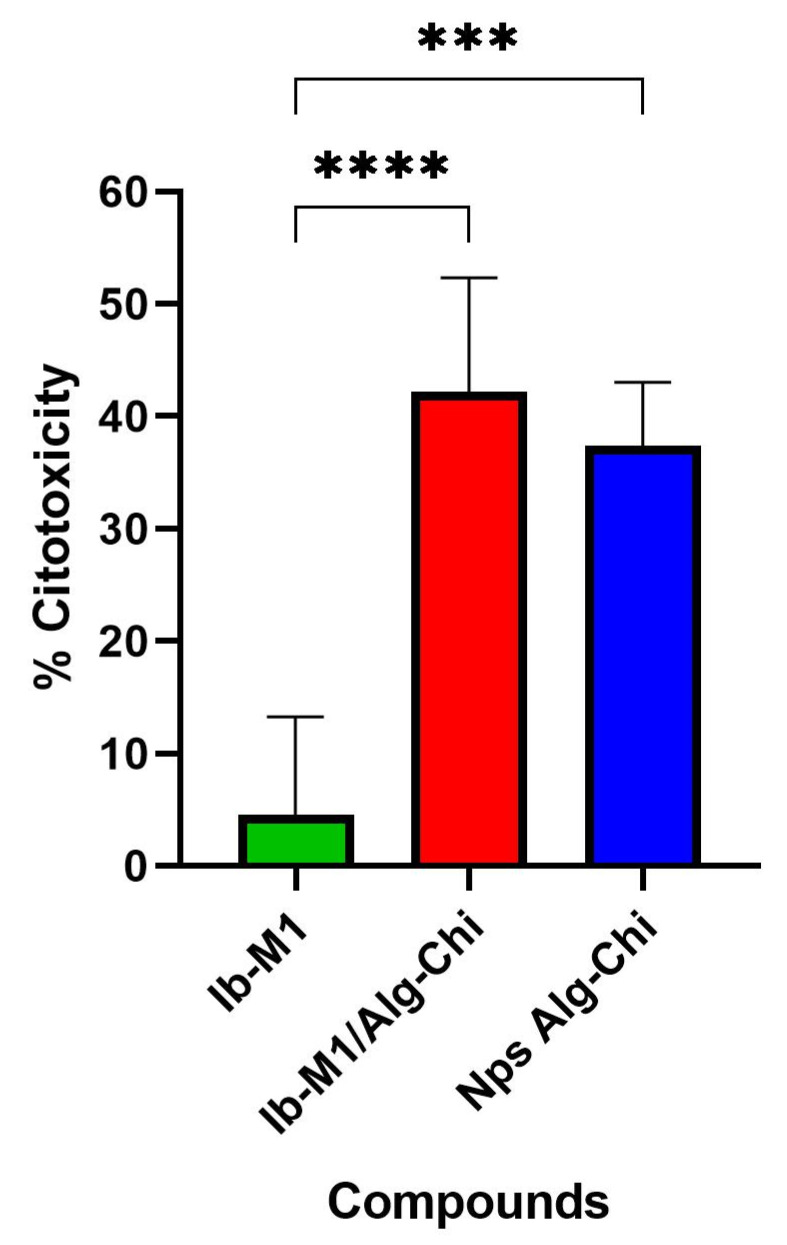
Cytotoxic activity of the free peptide Ib−M1 (12.5 µM), the bioconjugate Ib−M1/Alg−Chi (12.5 µM) and Alg−Chi NPs (0.4 mg/mL) in Vero cells. Each result was evaluated in triplicate in two independent experiments and expressed in terms of arithmetic average plus standard deviation. Cells maintained in culture medium without any treatment were used as a negative control (— Control; not shown). *p* < 0.0001: ****; *p* ≤ 0.0001: ***.

**Figure 7 polymers-14-03149-f007:**
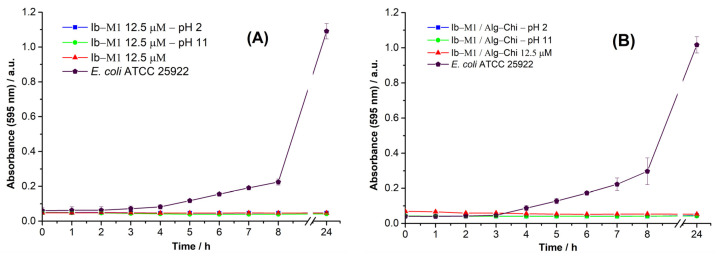
Effect of pH conditions on the antimicrobial activity of peptide Ib−M1 (**A**) and bioconjugate Ib−M1/Alg−Chi (**B**). Results of *E. coli* without the addition of compounds (⬟) are included for comparison purposes. Each result was evaluated in triplicate in two independent experiments and expressed in terms of arithmetic average plus standard deviation.

**Figure 8 polymers-14-03149-f008:**
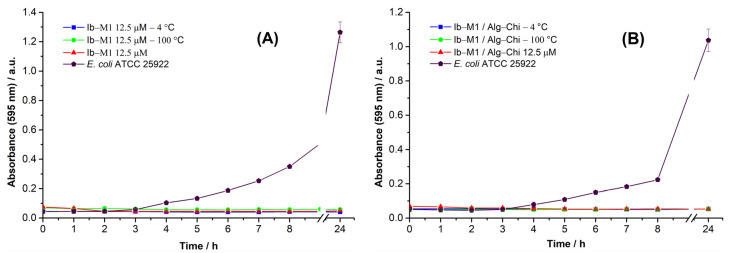
Effect of heat treatments on the antimicrobial activity of the Ib−M1 peptide (**A**) and the Ib−M1/Alg−Chi bioconjugate (**B**). Results of *E. coli* without the addition of compounds (⬟) are included for comparison purposes. Each result was evaluated in triplicate in two independent experiments and expressed in terms of arithmetic average plus standard deviation.

**Figure 9 polymers-14-03149-f009:**
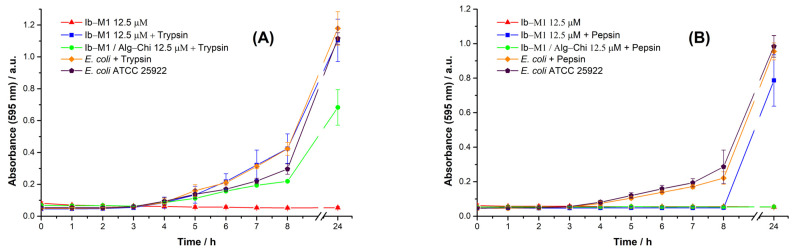
Antimicrobial activity of the peptide and the bioconjugate with prior exposure to trypsin (**A**) and pepsin (**B**). Results of *E. coli* without the addition of compounds (⬟) are included for comparison purposes. Each result was evaluated in triplicate in two independent experiments and expressed in terms of arithmetic mean plus standard deviation.

**Table 1 polymers-14-03149-t001:** Conditions used to evaluate the stability of peptides Ib−M1 and Ib−M1/Alg-Chi.

Condition	Features
pH	2 and 11
Temperature	4 and 100 °C
Proteases	Trypsin and Pepsin

## Data Availability

Not applicable.

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
