# Peer review of "Immobilization Systems of Antimicrobial Peptide Ib−M1 in Polymeric Nanoparticles Based on Alginate and Chitosan"

_polymers, 2022, doi:10.3390/polym14153149_

Round 1

Reviewer 1 Report

The manuscript entitles as “Immobilization systems of the antimicrobial peptide Ib-M1 in polymeric nanoparticles based on alginate and chitosan” is a research article, using an antimicrobial peptide with alginate/chitosan by ionic gelation as nanoparticles to against E. coli, it’s interesting, however need revision.

1.      Divide paragraphs into appropriate sections in the introduction, starting with the background of antimicrobial resistance, what kind of technique would overcome antimicrobial resistance, what antibacterial peptide would do, and then discussing nanoparticles, and then going back to what has already been done by others, but overall, the introduction gives no point or innovation of the study, instead presenting numerous details. I suggest, remove all the unrelated background and around the highlight/point of innovation, please arrange your background.

2.      Figure 2 Alg-Chi only, which kind of role the peptide and TPP play in the NPs.

3.      Please describe how peptides are chemically conjugated in lines 20-22 rather than using ionic gelation.

4.      Figure4, having trouble distinguishing the lines of the illustration, please make it more clear, and let us know which line represents which type of group.

5.      Figure 4 and Figure 5, in y-axis, what is the wavelength used for the absorbance? 

6.      Figure 6, The significance of each group should be labeled.

7.      Figure 7, Why would you use pH2, or pH7 to combat E coli, at this pH, most E coli would not grow.

8.      In the experiment, only one type of bacteria was examined, what about related bacteria?

9.      The peptide is an alki peptide. Please give us more information about the alki peptide's mechanisms of action, and peptide information, including its PI-value and other related to against microbial.

10.  After enzyme treatment of the peptide, would the function of the peptide be lost, and also the Chitosan also has an antibacterial effect, would not be able to see the control group of chitosan or alga-chi NPs.

11.  Please check the text properly as there are too many typos.

Author Response

Response to Reviewer 1

Comment 1: 1. Divide paragraphs into appropriate sections in the introduction, starting with the background of antimicrobial resistance, what kind of technique would overcome antimicrobial resistance, what antibacterial peptide would do, and then discussing nanoparticles, and then going back to what has already been done by others, but overall, the introduction gives no point or innovation of the study, instead presenting numerous details. I suggest, remove all the unrelated background and around the highlight/point of innovation, please arrange your background.

Response: We want to thank the reviewer for this comment. We have considered the comment and decided to change the introduction giving it another approach.

Comment 2: Figure 2 Alg-Chi only, which kind of role the peptide and TPP play in the NPs.

Response: We appreciate this comment and we have decided to include the analysis of the bands related to the TPP. Therefore, figure 2 was changed, including the value of the bands associated with TPP and improving the quality and size of the axes.

Figure 2 corresponds to the FITR spectrum of the nanoparticles. For this reason, an analysis of the role of the peptide cannot be included.

Comment 3. Please describe how peptides are chemically conjugated in lines 20-22 rather than using ionic gelation.

Response: We appreciate the reviewer's comment and fully agree with it, and we have reformulated this part in the manuscript.

Comment 4: Figure4, having trouble distinguishing the lines of the illustration, please make it more clear, and let us know which line represents which type of group.

Response: We appreciate this comment and fully agree with it; for this reason, figure 4 was modified. The highest and lowest concentrations were removed from the graph. Supplementary figure S1 was made. Figure S1.A shows the growth graphs of E. coli in the presence of the peptide at values between 0.78 and 6.25 µM. Figure S1.B shows the growth graphs of E. coli in presence the peptide at values between 12.5 and 100 µM.

Comment 5: Figure 4 and Figure 5, in y-axis, what is the wavelength used for the absorbance?

            Response: A wavelength of 595nm was used to perform the growth kinetics of E. coli. This data was included in graphs 4, 5,7, 8 and 9.

Comment 6: Figure 6, The significance of each group should be labeled.

            Response: We appreciate the reviewer comment. The data analysis was performed again, and the corresponding figure was constructed. Statistical values are indicated in the results description and in the figure 6.

Comment 7: Figure 7, Why would you use pH2, or pH7 to combat E coli, at this pH, most E coli would not grow.

            Response: The pH values 2 and 7 were used to determine the stability of the peptides. For this, a previous incubation of the peptides in glycine buffer at pH 2 and pH 11 was carried out. Later, the pH was adjusted again to 7, and the antimicrobial activity test against E. coli was performed. That is, when performing the antimicrobial activity tests, the pH used was 7.

Comment 8: In the experiment, only one type of bacteria was examined, ¿what about related bacteria?

            Response: We appreciate this comment. Only E. coli ATCC 25922 was used since it is the model microorganism used for evaluating antimicrobial compounds; also, the clinical importance of E. coli and its prevalence in different infections are recognized. However, it has already been shown in previous studies by our group that peptide Ib-M1 has activity against E. coli ATCC 45888 and isolated pediatric clinicians.

Comment 9: The peptide is an alki peptide. Please give us more information about the alki peptide's mechanisms of action, and peptide information, including its PI-value and other related to against microbial.

            Response: We appreciate the reviewer's comment, and we have reformulated this part in the introduction.

Comment 10: After enzyme treatment of the peptide, would the function of the peptide be lost, and also the Chitosan also has an antibacterial effect, would not be able to see the control group of chitosan or alga-chi NPs.

            Response: We would like to thank the Reviewer for this comment. The Nps Alg-Chi were evaluated at the same concentration of the bioconjugate, and it was shown that the Nps did not have an antibacterial effect on the microorganism.

Comment 11: Please check the text properly as there are too many typos.

            Response: We appreciate reviewer detection of these items. We have revised the document and unified the sources and we have made corrections to all of them.

Reviewer 2 Report

The manuscript is focused on the production and characterization of polymeric nanoparticles based on Alg and Chi by ionic gelation synthesize for immobilization the Ib-M1 peptide as an antibacterial agent. The idea is novel and the experimental design was generally well-done and well organized, and I recommend publication after the following corrections:

Abstract:

Line 16: Italicize the “E. coli”.

Introduction:

Lines 62-64: It is hard to understand “To achieve these strategies, alginate and chitosan nanoparticles (Alg-Chi Nps) have been used for supplying oral proteins and peptides, which have shown good alternatives, since they preserve their structure and, therefore, potentially their bioactivity”. Please rephrase it.

Line 74: It was claimed that “these polymers are the most used for the preparation of nanoparticles”. On what fact or reference?

Line 77: Please add an appropriate reference.

Line 94: “was evaluated” refers to what?

Materials and methods

Section 2.5: After the addition of 100 μL of bacterial inoculum to each well, the final concentration of the bacteria would be 5*104CFU/ml in 200 μL well. Is this according to the CLSI protocol?

Lien 224: r.p.m. or rpm? Please be constant throughout the text.

Table 1: Using “-“ between the pH and temperature values mislead the readers that the stability tests were performed in a range of pH and temperatures between 2-11 and 4-100 respectively. So, please revise.

Section 2.6.3: Here it is not mentioned what test was performed for analyzing peptide activity after enzyme treatment.

Line 243: replace “2.7” with 2.8.

 Line 244: it is recommended to use statistical methods for comparing the data obtained from antibacterial and Cytotoxicity tests. E.g., between peptide and immobilized peptide.

Results

Figure1: it is better to define abbreviations in figure captions, like Alg-Chi.

Lines 265-270: Based on what evidence the detected bands were assigned to different groups like carbonyl. So, some references should be mentioned for this section.

Figure 4: Due to the number of elements presented in this figure, it is not feasible to find some groups of elements. Please find a solution.

Figure 7: Replace “Algquit” with Alg-Chi in chart B title.

Figure 6: Is there any explanation for the large Standard deviation in the bioconjugate Ib-M1/Alg-Chi group?

Discussion

Lines 423-436: The reason why Ib-M1 is stable under the pH and temperature should be explained and discussed in this paragraph.

Author Response

Comment 1: Line 16: Italicize the “E. coli”.

Response: We appreciate your comment, and we have made the modification.

Comment 2: Lines 62-64: It is hard to understand “To achieve these strategies, alginate and chitosan nanoparticles (Alg-Chi Nps) have been used for supplying oral proteins and peptides, which have shown good alternatives, since they preserve their structure and, therefore, potentially their bioactivity”. Please rephrase it.

Response: We totally agree with your suggestion. We have modified the introduction taking into account the comments of reviewer 1 

Comment 3: Line 74: It was claimed that “these polymers are the most used for the preparation of nanoparticles”. On what fact or reference?

Response: We totally agree with your suggestion. We have modified the introduction taking into account the comments of reviewer 1 

Comment 4: Line 77: Please add an appropriate reference.

Response: We totally agree with your suggestion. We have modified the introduction taking into account the comments of reviewer 1 

Comment 5: Line 94: “was evaluated” refers to what?

Response: We totally agree with your suggestion. We have modified the introduction taking into account the comments of reviewer 1 

Comment 6: Section 2.5: After the addition of 100 μL of bacterial inoculum to each well, the final concentration of the bacteria would be 5*104CFU/ml in 200 μL well. Is this according to the CLSI protocol?

  Response: We appreciate the reviewer's comment. We have reformulated this part in the manuscript. Concentrations were specified according to CLSI.

Comment 7: Lien 224: r.p.m. or rpm? Please be constant throughout the text.

Response: We appreciate your suggestion, and we unified to rpm.

Comment 8: Table 1: Using “-“ between the pH and temperature values mislead the readers that the stability tests were performed in a range of pH and temperatures between 2-11 and 4-100 respectively. So, please revise.

Response: We totally agree with your suggestion. We have modified table 1, clarifying that the pH employees are 2 and 11.

Comment 9: Section 2.6.3: Here it is not mentioned what test was performed for analyzing peptide activity after enzyme treatment.

Response: We would like to thank the Reviewer for this comment. We have indicated at the end of section 2.7.3 that the antimicrobial activity was determined for the microdilution method.

Comment 10: Line 243: replace “2.7” with 2.8.

Response: Renumbered sections 2.7.1, 2.7.2, 2.7.3 and 2.8.

Comment 11:  Line 244: it is recommended to use statistical methods for comparing the data obtained from antibacterial and Cytotoxicity tests. E.g., between peptide and immobilized peptide.

Response: We appreciate this comment. Results of statistical analyzes of cytotoxicity were included.

Comment 12:  Figure 4: Due to the number of elements presented in this figure, it is not feasible to find some groups of elements. Please find a solution.

Response: We appreciate this comment and fully agree with it; for this reason, figure 4 was modified. The highest and lowest concentrations were removed from the graph. Supplementary figure S1 was made. Figure S1.A shows the growth graphs of E. coli in the presence of the peptide at values between 0.78 and 6.25 µM. Figure S1.B shows the growth graphs of E. coli in presence the peptide at values between 12.5 and 100 µM.

Comment 13:  Figure 7: Replace “Algquit” with Alg-Chi in chart B title.

Response: We appreciate this comment. The nomenclature of the compounds in the figure was unified.

Comment 14:  Figure 6: Is there any explanation for the large Standard deviation in the bioconjugate Ib-M1/Alg-Chi group?

Response: We appreciate this comment. Results of statistical analyzes of cytotoxicity were included.

Comment 15:  Lines 423-436: The reason why Ib-M1 is stable under the pH and temperature should be explained and discussed in this paragraph.

Response: We appreciate this comment. The characterization of the mechanism of action of this peptide is currently under investigation and will be revealed in future publications.

Round 2

Reviewer 1 Report

The manuscript entitles as “Immobilization systems of the antimicrobial peptide Ib-M1 in polymeric nanoparticles based on alginate and chitosan” is a revision article, it’s improve, but need revision.

1.      There is only one curve for FTIR, making it difficult to compare the efficiency of different materials.

2.      In terms of antimicrobials, authors only affect E coli, but what about the other microbial strains?

3.      In the supplymentary data, the author provides information on the antibacterial activity of the Ib-M1 peptide, additionally, chitosan has antibacterial properties, which increase or decrease when combined with peptides. Does it have a synergistic effect?

Author Response

Comment 1: There is only one curve for FTIR, making it difficult to compare the efficiency of different materials.

Response: We want to thank for this comment. The purpose of the FITR measurements was not to determine the efficiency of the materials. Instead, the objective was to corroborate the presence of alginate and chitosan in the nanoparticles obtained. Clarifications were included on page 3, paragraph 5, lines 122-125 and they are marked in red type.

Comment 2: In terms of antimicrobials, authors only affect E coli, but what about the other microbial strains?

Response: We appreciate this comment. Only E. coli ATCC 25922 was used since it is the model microorganism used for evaluating antimicrobial compounds; also, the clinical importance of E. coli and prevalence in different infections are recognized. However, it has already been shown in previous studies by our group that peptide Ib-M1 has activity against E. coli ATCC 45888 and isolated pediatric clinicians.

Some clarifications to this concern were included on page 2, paragraph 5, lines 84-85, and page 12, paragraph 4, lines 391-399. They are marked in red type.

Comment 3. In the supplymentary data, the author provides information on the antibacterial activity of the Ib-M1 peptide, additionally, chitosan has antibacterial properties, which increase or decrease when combined with peptides. Does it have a synergistic effect?

Response: We appreciate the reviewer's comment and fully agree with it.  We have included in the manuscript that a synergistic effect was observed in the combination of the Nps Alg-Chi with the peptide. Clarifications were included on page 9, paragraph 1, lines 306-312 and they are marked in red type.
